# Lysosomes in Cancer—At the Crossroad of Good and Evil

**DOI:** 10.3390/cells13050459

**Published:** 2024-03-05

**Authors:** Ida Eriksson, Karin Öllinger

**Affiliations:** Division of Cell Biology, Department of Biomedical and Clinical Sciences, Linköping University, 58185 Linköping, Sweden; karin.ollinger@liu.se

**Keywords:** lysosome, lysosomal positioning, exocytosis, extracellular vesicles, LMP

## Abstract

Although it has been known for decades that lysosomes are central for degradation and recycling in the cell, their pivotal role as nutrient sensing signaling hubs has recently become of central interest. Since lysosomes are highly dynamic and in constant change regarding content and intracellular position, fusion/fission events allow communication between organelles in the cell, as well as cell-to-cell communication via exocytosis of lysosomal content and release of extracellular vesicles. Lysosomes also mediate different forms of regulated cell death by permeabilization of the lysosomal membrane and release of their content to the cytosol. In cancer cells, lysosomal biogenesis and autophagy are increased to support the increased metabolism and allow growth even under nutrient- and oxygen-poor conditions. Tumor cells also induce exocytosis of lysosomal content to the extracellular space to promote invasion and metastasis. However, due to the enhanced lysosomal function, cancer cells are often more susceptible to lysosomal membrane permeabilization, providing an alternative strategy to induce cell death. This review summarizes the current knowledge of cancer-associated alterations in lysosomal structure and function and illustrates how lysosomal exocytosis and release of extracellular vesicles affect disease progression. We focus on functional differences depending on lysosomal localization and the regulation of intracellular transport, and lastly provide insight how new therapeutic strategies can exploit the power of the lysosome and improve cancer treatment.

## 1. Introduction

Lysosomes are small membrane-bound vesicles with a central role in the degradation of cellular macromolecules, waste material, and foreign particles. The degradation of worn-out biomolecules takes place in the lysosomal lumen, aided by acidic pH and around 60 different hydrolases [1,2]. The breakdown and recycling of internalized material creates new building blocks and sources of energy and places the lysosome as a cellular center for metabolic regulation. Since their discovery in the 1950s [3], the initial view of lysosomes as simple waste bags has been abandoned as more and more functions are ascribed to the organelle, and the lysosome is now considered a dynamic organelle that is essential for maintaining cellular homeostasis [4]. While autophagic, phagocytic and endocytic pathways are the main routes to lysosomal degradation, lysosomes also take part in various other processes, including cell signaling, nutrient sensing, cell death induction, cholesterol homeostasis, and immune signaling (Figure 1). In addition, by mediating fusion and fission events with other organelles, the number, size, and content of lysosomes can be controlled [5]. Due to its vital function for cellular homeostasis, lysosomal dysregulation can result in severe and sometimes fatal diseases [6]. Lysosomal exocytosis and the release of the content to the extracellular environment, either as soluble factors or as extracellular vesicles, participate in plasma membrane repair and cell signaling, and is considered a driving force of tumor progression by modulating the microenvironment to facilitate tumor spreading [7]. 

### 1.1. Lysosomal Characteristics

The lysosomal pH of around 4-5 provides an optimal environment for lysosomal enzyme activity and aids the degradation by reducing molecular interactions and denaturing proteins with various hydrolases [5,9]. Lysosomes harbor around 60 different enzymes, including lipases, nucleases, sulfatases, proteases, and peptidases, that together degrade most complex macromolecules. Cathepsins constitute the central family of proteases and are classified as either serine (cathepsin A and G), cysteine (cathepsin B, C, F, H, K, L, O, S, V, W, and X) or aspartyl (cathepsin D and E) proteases, according to the amino acid situated in the active site [10]. Cathepsins and other lysosomal hydrolases are synthesized as inactive precursors in the endoplasmic reticulum (ER). Most of the soluble lysosomal enzymes are tagged with mannose-6-phosphate (M6P) residues that are recognized by M6P-receptors in trans-Golgi for further transport to endosomes and lysosomes. Once delivered, the enzymes require proteolytic processing, either by other proteases or via autocatalysis, for activation. Generally, optimal activity also requires acidic pH [11,12,13], although some cathepsins, including cathepsin B, D, and L, can retain their proteolytic activity for several hours at a neutral pH [14,15]. Lysosomal enzymes that escape M6P-receptor binding are contained within vesicles and secreted extracellularly as inactive precursors. Once outside the cell, they are captured by M6P receptors on the plasma membrane and brought back to the lysosome via the endocytic pathway [16]. While most cathepsins are ubiquitous and widely distributed in high to moderate concentrations, others are more tissue specific [17]. Cathepsin-dependent degradation is crucial, not only for maintaining cell homeostasis, but also to control cell growth and development by regulating the levels of hormones and growth factors. Furthermore, cathepsins are involved in the adaptive immune response by processing antigens for presentation by MHC-II molecules, and play roles in inflammation and immune cell migration by regulating the activation and function of immune cells [15]. Cathepsin deficiency results in the accumulation of undigested proteins, which leads to impaired lysosomal function and can cause severe embryonic and post-natal abnormalities, including neurodegeneration, skeletal defects, and cardiomyopathy [6,18,19,20]. Although mainly localized within the lysosomal lumen, cathepsins can function at other cellular locations. Upon lysosomal damage, cathepsins released to the cytosol participate in cell death signaling [21,22], while a nuclear localization can modulate gene transcription and proliferation [23]. Moreover, cathepsins secreted extracellularly take part in, for example bone remodeling, degradation of the extracellular matrix (ECM), and the shedding of receptors and adhesion molecules [24]. 

#### 1.1.1. The Lysosomal Membrane

Lysosomes are surrounded by a phospholipid bilayer containing a high number of glycosylated membrane proteins, which provide the glycocalyx—a continuous coat of polysaccharides on the luminal side. This layer is important for protecting the membrane from the action of the hydrolytic enzymes present in the lysosomal lumen [25,26,27]. Previously considered to mainly separate the acidic lysosomal lumen from the cytoplasm, proteomics and functional studies have concluded that the lysosomal membrane participates in numerous cellular processes such as membrane fusion, signaling, and molecular transport [28,29,30]. The lysosomal membrane is estimated to harbor over 250 different lysosomal membrane proteins with diverse functions, including ion and metabolite transporters, and factors important for membrane tethering and lysosomal positioning (Figure 2) [5,28,31]. Interestingly, several membrane proteins can have dual functions. While the transmembrane part controls one process, for example transport across the membrane, the cytosolic part is involved in organelle contact or signaling with other compartments. 

Lysosomal-associated membrane proteins (LAMP)-1 and -2 are the most abundant proteins in the lysosomal membrane, constituting around 50% of all lysosomal membrane proteins. A functional redundancy has been observed, as deficiency in either LAMP1 or LAMP2 produce surviving offspring, while double deficiency results in embryonic death [32]. Accordingly, sharing a 34% amino acid homology, both proteins harbor a large N-terminal luminal domain, a single membrane-spanning region, and a short C-terminal cytoplasmic tail, playing roles in various crucial cellular functions [33]. LAMP1 is a key mediator of lysosomal docking to the plasma membrane to allow lysosomal fusion and subsequent exocytosis of lysosomal content [34]. A specific LAMP2 isoform, LAMP2a, acts as an autophagy receptor, discussed in more detail below. Both LAMP1 and LAMP2 function in cholesterol regulation as they bind free cholesterol inside the lysosomal lumen and interact with the transmembrane and soluble cholesterol exporters, Niemann Pick type C protein (NPC) 1 and 2 [35]. LAMP2 and lysosomal integral membrane protein (LIMP) 2 have also been shown to facilitate direct cholesterol export [35,36]. LIMP2 is also known as a transporter for β-glucocerebrosidase [37]. In addition, LAMP proteins, as well as NPC1 and the lysosome-enriched tetraspanin CD63, regulate lysosomal motility and fusion with other organelles [31,38,39,40]. The acidic pH in the lysosomal lumen is maintained by the vacuolar H^+^-ATPase (V-ATPase), a large protein complex that imports protons at the expense of ATP [41]. The proton gradient is essential for the efflux of ions and small degradation products out of the lysosome, since it facilitates proton-driven transport of ions, amino acids, sugars, and other metabolites via H^+^ coupled co-transporters [42]. In addition to acidification, the transmembrane part of the ATPase protein can form complexes in the opposing membranes during membrane fission events and participates in membrane fusion during synaptic vesicle release [43,44]. Membrane proteins associated with lysosomal transport and membrane fusion are further described below and in Section 2 and Section 3. For a more thorough review of lysosomal membrane proteins and their function, please refer to [27,45].

#### 1.1.2. Calcium Signaling

Intracellular calcium (Ca^2+^) signaling relies on a continuous release of Ca^2+^ to the cytosol and subsequent re-accumulation into storage organelles. The resting concentration of Ca^2+^ is approximately 100 nM in the cytosol but can increase to 0.5–1 µM upon stimulation. Although most of the intracellular Ca^2+^ is stored in the membranous network of the ER, lysosomes also have Ca^2+^ storing properties [46]. The Ca^2+^ concentration has been estimated to 400-600 µM in lysosomes, which is close to the concentration found in the ER [47,48]. Ca^2+^ signaling from the lysosome regulates several processes, such as membrane trafficking, autophagic recycling, and communication between organelles, and is required for lysosomal fusion with other membranous compartments. The main types of lysosomal Ca^2+^ permeable cation channels are the transient receptor potential mucolipin (TRPML1) channel, the two-pore channels (TPC) TPC1 and TPC2, and the trimeric two transmembrane-spanning channel P2X4. Other Ca^2+^ channels have been identified as well, but their function is not yet confirmed [49]. Uptake of Ca^2+^ into the lysosome is thought to be driven by the proton gradient and occur via a Ca^2+^/H^+^ exchanger or a Ca^2+^ transporter. The channels are regulated in several ways, mainly via the binding of small molecules, such as nicotinic acid adenine dinucleotide phosphate and ATP, but also by alterations in pH, changes in nutritional status, and other cellular stresses and stimuli [50,51]. TRPML1 is the most studied lysosomal cation channel and regulates fusion events between lysosomes and other cellular organelles. A TRPML1-mediated Ca^2+^ release is also needed for lysosomal biogenesis, lysosome reformation, and exocytosis [52].

### 1.2. Lysosomal Degradation

Cargo destined for lysosomal degradation is delivered to lysosomes via two major pathways: endocytosis and autophagy. Autophagy, meaning self-eating, refers to the degradation and recycling of unnecessary or dysfunctional intracellular components via a lysosome-dependent mechanism [53]. It is induced as a survival strategy under nutrient-deficient conditions, but can also be activated in physiological processes, including embryonic development, cell differentiation, the regulation of immune cells, and the elimination of intracellular microbes. 

Autophagy occurs via three major routes: macroautophagy, chaperone-mediated autophagy (CMA), and microautophagy (Figure 3). In macroautophagy, a double membrane, known as the phagophore, is formed around the cytoplasmic material selected for degradation. The generated autophagosome then fuses with a lysosome, creating the autolysosome, in which degradation of the sequestered material occurs [54,55]. Autophagy serves both as a nonselective process, activated during starvation to provide new nutrients, or as a selective event, during which ubiquitin-tagged proteins or organelles are targeted by autophagy receptors for degradation [56]. Larger structures, such as damaged cell organelles, are eliminated via macroautophagy, while CMA and microautophagy mainly manage proteins or smaller organelle structures [57,58]. CMA is a selective form of autophagy, where cytosolic proteins bearing the specific pentapeptide target motif KFERQ are recognized by the cytosolic chaperone HSC70. HSC70 binds to the motif and brings the target protein to the lysosomal surface by associating with the lysosomal membrane receptor LAMP2a. This induces the formation of a LAMP2a multimeric complex and allows translocation of the substrate proteins into the lysosomal lumen [57]. During microautophagy, invagination of the lysosomal membrane will provide direct engulfment of cytosolic material, a process that is mainly considered non-selective [58]. However, a selective form of microautophagy, which occurs solely in late endosomes, has also been described [59]. During endosomal microautophagy, HSC70 targets KFERQ-like motifs on the substrate protein, much like the process in CMA. Instead of binding to lysosomal LAMP2a, HSC70 interacts with acidic phospholipids in the endosomal membrane to facilitate substrate delivery into the organelle. 

### 1.3. Endolysosomal Maturation and Lysosome Biogenesis

Extracellular material is delivered to the lysosome via the endosomal pathway (Figure 4). Material endocytosed at the plasma membrane is first brought to the slightly acidic early endosomes, where cell surface receptors are dissociated from their ligands [60]. Most of the material taken up by endocytosis is recycled back to the plasma membrane via recycling endosomes [61]. Cargo destined for degradation continues down the endocytic route, where early endosomes mature into late endosomes and ultimately lysosomes [5,60]. Maturation from early to late endosomes involves the generation of intraluminal vesicles, formed by invaginations of the endosomal limiting membrane, which allows for efficient sorting of the transmembrane cargo between the limiting membrane and intraluminal vesicles [60]. Due to the accumulation of intraluminal vesicles, late endosomes are called multivesicular bodies or multivesicular endosomes (MVEs). The vesicles also move from the cell periphery to the perinuclear area, a transport that is dependent on Rab proteins. Rab5 facilitates the transport of early endosomes, but is then replaced by Rab7 to manage late endosomal trafficking [62]. Maturing endosomes gradually acquire degradative capacity via delivery of components from the trans-Golgi network, either via direct vesicle transport or endocytic uptake from the secretory pathway [5]. Combined with transient and complete fusion events between late endosomes and pre-existing lysosomes, the level of lysosomal hydrolases, membrane proteins, and proton pumps are increased to facilitate degradation of the cargo [5,60]. 

### 1.4. Nutrient Sensing and Transcriptional Regulation of Lysosomal Biogenesis

Lysosomal biogenesis is controlled by the transcription factor EB (TFEB) and other members of the MiT/TFE family of transcription factors, including TFE3 and MITF [63,64]. Promotors targeted by these transcription factors contain a 10-base E-box-like motif, the CLEAR (coordinated lysosomal expression and regulation) element, which is found in many genes regulating lysosomal function, and autophagy [65]. MiT/TFE activity is controlled by both post-translational modifications and protein/protein interactions. The phosphorylation of serine residues governs the intracellular localization and activity of the transcription factors. The most important regulator of MiT/TFE activity is the mechanistic target of rapamycin (mTOR), a serine/threonine kinase that is part of the mTOR complex 1 (mTORC1) [66]. Under nutrient-rich conditions, mTORC1-mediated phosphorylation of MiT/TFE proteins induces their binding to 14-3-3 proteins and cytosolic retention [67,68]. Upon starvation or lysosomal stress, mTORC1 inhibition releases the transcription factors from the 14-3-3 proteins. In concert, a local increase in Ca^2+^, released via lysosomal TRPML1 activates the Ca^2+^-dependent phosphatase calcineurin. Subsequent dephosphorylation of MiT/TFE proteins allows their nuclear translocation and activation of autophagy [69,70]. When nutrients are available again, TRPML1 depletion reduces Ca^2+^ release and calcineurin activation, resulting in a decreased activation of MiT/TFE transcription factors and inhibition of autophagy. While mTORC1 is the main negative regulator of MiT/TFE activity, other kinases such as AKT, ERK2, and GSK-3 can induce their cytosolic retention as well [50,71]. Furthermore, the dephosphorylation of protein phosphatase 2 (PPA2) has been shown to induce nuclear translocation of TFEB and TFE3 under oxidative stress [72]. Recently, an overlap between CLEAR motifs recognized by MiT/TFE and the E-box motifs recognized by the c-MYC transcription factor was found [73]. By competing with the MIT/TFE transcription factors for the binding sites to the promotor regions, c-MYC reduces lysosomal biogenesis. This rheostat mechanism is epigenetically controlled by histone acetylation and deacetylation and modulates the balance between lysosomal biogenesis and cell proliferation.

### 1.5. Lysosomal Involvement in Regulated Cell Death

Already in the mid-1950s, DeDuve realized that the rupture of the lysosomal membrane and ensuing release of its lytic content into the cytosol would cause cell death [3], a mechanism called lysosomal membrane permeabilization (LMP). Since then, the view of lysosomal participation in cell death has expanded considerably. The cellular reaction is finetuned and dependent on the degree of membrane destabilization; from stress-mediated repair activated by minor damage, via different forms of regulated cell death induced by intermediate leakage, to total cell lysis upon lysosomes rupture [74,75]. The mechanism of LMP is not completely elucidated, but it is evident that several different mechanisms can take part. Lysosomal membrane damage can be inflicted by a variety of internal and external stimuli, including free radicals, lysosomotropic drugs, endogenous pore-forming proteins, and an accumulation of sphingomyelin and protein aggregates [76,77,78,79,80]. Lysosomal disruption is also the source of cellular entry for several different viral and bacterial toxins [81,82].

Cathepsins are the main executioners of lysosome-dependent cell death (Figure 5). Depending on experimental system and LMP-inducer, cathepsins can function as both triggers and enhancers of cell death mechanisms. LMP can occur upstream of the mitochondrial pathway [83,84,85,86], where inhibition of cathepsin D and cysteine cathepsins attenuates cell death [87,88,89,90]. Cathepsins released to the cytosol promote apoptosis by affecting members of the apoptosis-regulating Bcl2 family. By cleaving the pro-apoptotic protein Bid into its active truncated form, tBid, cathepsins can trigger cytochrome c release via Bax and Bak oligomerization [91,92,93]. In addition, the degradation of anti-apoptotic proteins such as Bcl-2, Bcl-XL, and Mcl-1 [89] allows Bax/Bak oligomerization and cytochrome c release from mitochondria. Cathepsins can also amplify apoptosis signaling downstream of the mitochondrial cytochrome c release during growth factor deprivation, ultraviolet radiation, and death receptor activation (Figure 5) [94,95].

Lysosomal damage is likewise involved in cell death with necrotic morphology, where LMP can act as a proteolytic amplifier to allow the disintegration of cellular organelles [96]. Previously, necrosis was described as purely accidental, implicated after massive cell damage and resulting in cell lysis, but recent research has identified several mechanisms of regulated cell death with necrotic-like morphology [97]. LMP and massive release of cathepsins have been demonstrated during necroptosis, where inhibition of cathepsins B and D reduces cell death [98,99]. Pyroptosis is a variant of regulated cell death that is activated by the innate immune system as an inflammatory defense mechanism, where LMP and release of cathepsins are important mediators. During pyroptosis, the assembly of the inflammasome protein complex activates pro-caspase-1 and induces an inflammatory response via the activation of interleukin-1β and interleukin-18. ROS, generated during phagocytosis of, for example, oxidized LDL particles or neurotoxic aggregates, cause LMP and ensuing release of cysteine cathepsins, which induce NLRP3 inflammasome activation (Figure 5) [100].

Although autophagy is primarily considered a survival strategy, excessive autophagy can lead to cell death. In many instances, autophagy coincides with the induction of other forms of regulated cell death, but a distinct death routine has been defined as autophagy dependent cell death (ADCD) [101]. The main criteria for ADCD are that cell death is dependent on at least two proteins of the autophagic machinery and that the process is reversible via genetic of pharmacologic intervention [97,102]. Whether there are differences in the mechanistic regulation between the autophagic process promoting survival or inducing cell death is not yet elucidated. ADCD has been associated with excessive engulfment of cytoplasmic material during mitophagy and ER-phagy [102]. Studies have shown participation of LMP in ADCD, where exaggerated autophagy results in the accumulation of cholesterol and ceramide in the lysosomes, which eventually cause destabilization of the membrane and release of lysosomal content to the cytosol [103,104,105].

## 2. Lysosomal Positioning

In non-polarized cells, most lysosomes are located in a perinuclear cluster adjacent to the microtubule-organizing center (MTOC) [106,107]. Clustering of lysosomes is however not defined to the perinuclear area, but occurs throughout the intracellular space, and is thought to facilitate the interaction with other lysosomes and organelles [108]. In polarized cells such as neurons, lysosomes are found in the whole cytoplasmic compartment, even if they are most abundant in the neuronal cell body and are more sparsely dispersed in dendrites and axons [109]. Recent studies have shown that lysosomes have different functions and heterogeneous properties depending on their intracellular localization. Perinuclear lysosomes are relatively immobile and have a low intraluminal pH compared to lysosomes residing in the cell periphery, which are more dynamic, albeit with a reduced acidity [110]. Due to their diverse characteristics, different lysosomal populations can have separate functions. Peripheral lysosomes are suggested to constitute the main subset to participate in lysosomal exocytosis and plasma membrane repair [111]. This population has a higher pH and contains inactive hydrolases [112]. In addition, the peripheral lysosomes control mTORC1 activation [106]. During nutrient-rich conditions, the presence of amino acids stimulates anterograde transport of lysosomes towards the plasma membrane, where growth factors induce mTORC1 activation via an Akt-dependent signaling pathway [106,113]. Upon starvation, lysosomes are relocated to the perinuclear area, which stimulates autophagosome/lysosome fusion to increase autophagic flux and nutrient availability [106,114].

### 2.1. Regulation of Lysosomal Transport

Lysosomes move bidirectionally along the cytoskeleton in order to exchange material and allow intracellular communication [50]. Shorter transport occurs along actin microfilaments and is relatively slow (≈0.1 µm/s), while long distance transport is faster (≈1 µm/s) and takes place on microtubule tracks [115]. Microtubule transport is orchestrated by dynein and kinesin motor proteins [116,117]. In non-polarized cells, the microtubule plus-ends project towards the cell periphery and the minus-ends towards the MTOC [118]. In neurons, the microtubule plus-ends always point toward the axon terminal, but can have a mixed orientation in the dendrites, and consequently, kinesins and dyneins can mediate both anterograde and retrograde transport [119,120]. While there are only two dyneins responsible for minus-end directed transport, the mammalian genome encodes 45 proteins belonging to the kinesin superfamily (KIFs) of proteins. Even more variants can then be created by alternative mRNA splicing [121]. Kinesins are classified as N-kinesins, C-kinesins, and M-kinesins, representing the positioning of the motor domain near the N-terminal, near the C-terminal, or in the middle [122]. N-kinesins (kinesin-1 to 12) are the main subset responsible for plus-end directed transport, while C-kinesins (kinesin-14) facilitate minus-end directed trafficking and M-kinesins (kinesin-13) mediate depolymerization of microtubule [123]. 

### 2.2. Anterograde Transport

All kinesin proteins have a motor domain that attaches to the microtubule and drives the transport through ATP hydrolysis, and a tail domain that interacts with adaptor proteins [121]. Lysosomal movement is regulated by several kinesins, including kinesin-1 (KIF5A, KIF5B and KIF5C) [124,125,126], kinesin-2 (KIF3) [127], kinesin-3 (KIF1A and KIF1B) [128,129], and kinesin-13 (KIF2) families [130], where kinesin-1 is the best characterized. Why the need for such variety of kinesins is not fully elucidated, but different kinesins have been shown to regulate the transport of lysosomes along various microtubule tracks. While KIF5B is preferentially attached to perinuclear, acetylated tubulins, KIF1A prefer peripheral tyrosinated tubulins [131]. In neurons, kinesins utilize microtubule in different cellular compartments. Kinesin-1 motor proteins are selective to axonal microtubules, while members in the kinesin-3 family facilitate transport in both axons and dendrites [132,133,134]. 

Kinesin interaction with lysosomes is mediated by small GTPases, various effector proteins, and lipids (Figure 6). The coupling of kinesin-1 and kinesin-3 to lysosomes is facilitated by the multisubunit complex, BORC [131,135,136]. The association of BORC to lysosomes recruits the GTPase Arl8 and its effector SKIP and links lysosomes to the kinesin motor proteins [126,131,135,136]. Alternatively, the ER membrane protein protrudin can form contact sites with lysosomes by interacting with Rab7 and phosphatidylinositol 3-phosphate (PI_3_P) in the lysosomal membrane. This enables the transfer of kinesin-1 from protrudin to the motor adaptor protein FYCO1 [137], which is initiated by amino acid-stimulated production of PI_3_P, mediated by VSP34 [113].

### 2.3. Retrograde Transport

Dynein-mediated regulation of retrograde transport acts via a similar mechanism as kinesins, with the exception that only one dynein is involved in lysosomal transport. There are two different types of dyneins; cytoplasmic dynein and axonemal dynein. Axonemal dynein is responsible for transport within flagella and cilia, while cytoplasmic dynein drives the majority of organelle transport towards microtubule minus-ends in the cell [138]. As for kinesins, the transport is mainly one-way directed in non-polarized cells, and the organelles are transported from the cell periphery to the cell center [139]. Dynein is a multimeric protein complex, which interacts with another multisubunit complex, dynactin, to attach lysosomes to the microtubule [140,141]. Recruitment of the dynein/dynactin complex to lysosomes can be regulated via several mechanisms (Figure 6). The lysosome-associated GTPase Rab7 is localized to late endosomes and lysosomes in an active GTP-bound state, and recruits the effector protein Rab7-interacting lysosomal protein (RILP) to couple the organelle to the dynein/dynactin complex [142]. Alternatively, local Ca^2+^ release via TRPML1 activates the lysosomal Ca^2+^-sensor ALG2, which interacts with dynein/dynactin to mediate the retrograde transport of lysosomes [114]. Upon starvation, this transport is induced by the inactivation of mTORC1 and succeeding activation of TFEB-induced transcription of TRPML1 [114,143]. Next, the lysosomal membrane protein TMEM55B can directly interact with the motor adaptor protein, JIP4, to recruit the dynein/dynactin complex, which is also regulated by starvation-induced TFEB transcription [144,145]. In addition to TMEM55B, other lysosomal membrane proteins such as LAMP1 and LAMP2 have been shown to facilitate retrograde transport of lysosomes, either by direct coupling with dynein/dynactin, or via an unknown adaptor [31]. In addition, retrograde transport of lysosomes can be facilitated by kinesin-14, the only kinesin that orchestrates minus-end directed transport [146]. 

## 3. Secretion from the Lysosomal Pathway 

Secretion is the regulated release of intracellular soluble proteins, vesicles, or vesicular content to the extracellular space. Initially, classical or conventional secretion was defined for the release of, for example, hormones and neurotransmitters. In this route, secretory proteins containing a signaling peptide are transported from the ER to the Golgi apparatus, packed into secretory vesicles and subsequently released upon vesicle fusion with the plasma membrane. In the 1990’s, unconventional protein secretion (UPS) emerged, which includes the release of leaderless cargo, i.e., content-lacking signaling peptides. Overall, four different types of UPS are recognized: transport through pores formed in the plasma membrane (type I), secretion via ATP-binding cassette (ABC) transporters (type II), secretion in vesicles of autophagosomal/endosomal origin (type III), and direct transport from the ER to the plasma membrane, bypassing Golgi (type IV) [147]. Belonging to UPS type III, lysosomal exocytosis involves the release of mature lysosomal content outside the cell (Figure 7) [148]. While initially thought to be limited to specialized secretory hematopoietic cells, it is now recognized as a ubiquitous event occurring in all types of cells [149,150,151]. Further, autophagosomes can exocytose and release the content extracellularly as an alternative to lysosomal degradation [148,152]. During recent years, small extracellular vesicles, exosomes, that originate from MVEs of the endosomal system, has been identified as important mediators of intercellular communication [153].

### 3.1. Release of Extracellular Vesicles

Extracellular vesicles (EVs) are heterogenous, membrane-limited particles released by cells to the extracellular environment. They play crucial roles in intercellular communication and are involved in various signaling processes allowing cells to exchange proteins, lipids, and genetic material [154]. The two main types of EVs are exosomes and ectosomes, each with distinct biogenesis and characteristics. In this review, the term EV is used to include both exosomes and ectosomes, or when the origin of vesicles is not known or stated. Exosomes (30–150 nm in diameter) originate from MVEs in the endosomal system. As previously mentioned, intraluminal vesicles are produced by the inward budding of the MVE-limiting membrane. The MVEs can then take different routes; either they fuse directly with lysosomes or autophagosomes to allow cargo degradation [155], or they relocate to the plasma membrane and secrete the intraluminal vesicles as exosomes (Figure 7). It remains largely unexplored how the balance between the degradative and secretory capacity is regulated, and the MVEs can remain in the cytosol for different periods of time [156]. 

The biogenesis of exosomes in MVEs includes cargo selection and targeting, followed by the formation of membrane invaginations. After this, scission of the invaginations allows the uptake of the cargo inside intraluminal vesicles. It is a complex multistep process that often involves the sequential recruitment of proteins belonging to the endosomal complex required for transport (ESCRT) I-III; although, ESCRT-independent mechanisms involving ceramides and tetraspanins have been identified as well [157,158,159]. During exocytosis, MVEs are transported to the plasma membrane by interacting with cytoskeletal actin and microtubules. Several members of the Rab GTPase family, as well as actin-binding proteins such as cortactin, are involved in the transport and docking to the plasma membrane [160,161,162]. Fusion steps are then coordinated by SNARE proteins and different small GTPases, such as Ral-1, which are active in the fusion of MVEs with the plasma membrane and the ensuing exosome release [154,163]. 

Ectosomes, also called microvesicles or shedding vesicles, are formed by outward budding from the plasma membrane [149,164]. Compared to exosomes, they are larger, usually 50–500 nm in diameter (and up to over 1000 nm) and include vesicles of different origin and of variable chemical composition and signaling abilities. The ectosomes are named based on their origin (cell type), size, morphology, and cargo content, and include, for example, apoptotic bodies, oncosomes [165], and migrasomes [164,166]. Several steps in the formation of ectosomes are similar to those described for the generation of exosomes. The selection and targeting of cargo occurs at the plasma membrane and the biogenesis of the vesicles requires, apart from increase in Ca^2+^ levels, induction of membrane phospholipid asymmetry, altered cholesterol levels, and rearrangement of the actin cytoskeleton [167]. The mechanistic details of the biogenesis of EVs remains, however, to be elucidated in detail [164]. 

Although autophagy mainly is defined as a recycling process, secretory autophagy is possible. It is especially used for the disposal of toxic proteins, immune signaling, and pathogen surveillance [148]. For example, pathogens can be released extracellularly when degradation fails, and some viruses can utilize secretory autophagy to exit cells. The secretion of antimicrobial molecules, cytokines, etc., can also be used to induce an immune response. Studies have shown that secretory autophagy does not only represent a mechanism for immune surveillance but allows extracellular release of specific signaling molecules and membrane transporters as well [168]. Autophagosomes can fuse directly with the plasma membrane to release soluble proteins and vesicles with mature autophagosomal content. Alternatively, they can fuse with MVEs, forming a hybrid organelle called the amphisome. Secretion from amphisomes includes exosomes with autophagosome/exosome content and autophagic degradation products (Figure 7) [169]. 

### 3.2. Lysosomal Exocytosis

Lysosomal exocytosis is involved in several widespread functions such as plasma membrane repair, cell communication, antigen presentation, and bone resorption [170]. It is also recognized as an alternative way to eliminate cellular waste [171], and has been implicated in the release of ATP in the CNS [172] and as a response to oxidative stress [173]. All lysosomal exocytosis events are Ca^2+^ regulated and respond to increased intracellular free Ca^2+^ [174], either via influx from the extracellular environment or release from intracellular Ca^2+^ stores [175,176]. Interestingly, Jaiswal et al. showed that an increase in intracellular Ca^2+^ by the employment of a Ca^2+^ ionophore only stimulated exocytosis of mature lysosomes and had no effect on the exocytosis of post-Golgi vesicles or early and late endosomes [177]. 

An overexpression of TFEB has been found to increase lysosomal trafficking toward the cell periphery and promote fusion with the plasma membrane. Among TFEB-regulated genes, the lysosomal cation channel TRPML1 was found responsible by rising intracellular Ca^2+^ [176,178]. The increased Ca^2+^ levels activate synaptotagmin VII, which plays an important role in lysosome exocytosis [179]. By interacting with the lysosomal v-SNARE VAMP7, and the t-SNAREs SNAP-23 and syntaxin 4 on the plasma membrane, synaptotagmin VII controls membrane fusion (Figure 8) [180]. Lysosomal exocytosis is also dependent on LAMP1, and knockdown of this protein inhibits the docking of lysosomes to the plasma membrane [34]. Lysosome fusion with the plasma membrane will result in the appearance of lysosomal membrane proteins at the plasma membrane, and the detection of the luminal part of LAMP1 at the outer leaflet is often used as markers of lysosomal exocytosis [181]. 

### 3.3. Lysosome-Mediated Plasma Membrane Repair

Damage to the plasma membrane disrupts cellular integrity and is an acute threat to cell survival. It can be evoked by several causes, such as physical injury, mechanical stress, free radicals, or exposure to toxins from bacteria. Membrane lesions cause a Ca^2+^ influx from the extracellular environment, which triggers resealing of the plasma membrane within seconds after the damage [182]. Plasma membrane resealing is mediated by the donation of intracellular membranes, and Rodriguez et al. were the first to identify lysosomal exocytosis and fusion with the plasma membrane as a repair mechanism [174]. By donating its own membrane, the lysosome forms a patch over the lesion. Studies of plasma membrane damage, inflicted by invasion of the protozoa *Trypanosoma cruzi*, reveal that the release of acidic sphingomyelinase (ASMase) from the lysosome promotes remodeling of the outer leaflet of the plasma membrane and stimulates wound repair [183]. In addition, extracellular release of lysosomal proteases contributes to the repair mechanism, where proteolysis by cathepsins B and L promotes more efficient membrane access to ASMase, while cathepsin D facilitates ASMase inactivation and wound removal [184]. Recruitment of autophagy-related key proteins, such as LC3-II and ATG5, contributes to lysosome-mediated plasma membrane repair [185], and recent research has demonstrated the importance of Arl8b for the repair [186]. Moreover, a screening using a lentiviral shRNA library identified Rab3a and Rab10 as crucial mediators of plasma membrane repair [111]. 

After membrane damage and lysosomal repair, the membrane lesion must be removed to restore plasma membrane function, and several studies have identified endocytic internalization as the compensatory mechanism [187,188]. Do Couto et al. presented evidence that LAMP2 is an important player in the endocytosis process. LAMP2 deficiency leads to an increased cholesterol accumulation. Exocytosis of lysosomes enriched in cholesterol impairs the caveolin-1 distribution at the cell surface and prevents endocytosis. Thus, LAMP2 is crucial for the ability to perform endocytosis of lysosome-derived membrane parts [189]. Alternatively, the plasma membrane patch is removed by shedding the membrane through the generation of ectosomes. We recently showed that irradiation with UVA caused plasma membrane damage and lysosomal exocytosis. In melanocytes transfected with GFP-LAMP1, the UVA-induced plasma membrane damage was followed by the generation of LAMP1-positive ectosomes, showing that the shedded membrane was of lysosomal origin [190]. The mechanism controlling the shedding of the lysosome-derived membrane parts is not known. One possibility is that components released extracellularly upon lysosomal exocytosis are involved. Interestingly, Wang et al. recently showed that EGF signaling could facilitate the generation of ectosomes by activating the Rho family small G protein Cdc42, and at the same time, EGF signaling blocked EGFR endocytosis [191]. However, this remains to be proven and several other signaling pathways are possible [192].

## 4. Lysosomal Involvement in Cancer 

Due to its essential role in cellular homeostasis, the lysosomal system is often hijacked in cancer diseases, where malignant progression is promoted by altered metabolism and enhanced lysosomal exocytosis. The identification of TFEB as the master regulator of lysosomal activity, and the fact that TFEB and other members of the MiT/TFE family are considered oncogenes in several cancers, have highlighted the importance of lysosomes in cancer and underscored its potential as a therapeutic target [50]. Malignant transformation is associated with a gradual acquisition of proliferative, migratory, and invasive properties to enable tumor growth and spreading to distant locations. Lysosomes can contribute to this progression via various mechanisms, discussed in more detail below. 

### 4.1. Autophagic Rewiring

Cancer cells need to alter their metabolism to adapt to an increased proliferation rate and necessity to survive and grow in nutrient poor and hypoxic conditions. Therefore, the lysosomal system is often dysregulated [193]. A high expression of lysosomal proteases is associated with cancer progression and a poor prognosis in several types of cancer, including breast cancer, colorectal cancer, lung cancer, ovarian cancer, and pancreatic cancer [194]. Not surprisingly, cancer cells often display larger and more active lysosomes [195,196,197]. Although these changes often are correlated to a high risk of disease recurrence and poor prognosis, induction of autophagy and increased lysosomal function can have both tumor-suppressive and tumor-promoting effects. Early during tumor transformation, autophagic degradation helps to remove damaged proteins and organelles to prevent tumor initiation. However, later in the disease, autophagy functions as a cancer promotor [198,199]. Activation of autophagy is often seen in hypoxic regions of tumors, where poor vascularization causes a lack of nutrients and oxygen. Cancer cells are thus able to survive in poor conditions by utilizing autophagic recycling to provide energy. Inhibition of autophagic pathways, as well as the prevention of proteasome-dependent degradation, sensitizes tumor cells to metabolic stress, demonstrating its importance for maintaining cellular homeostasis in cancer [103,104,200,201]. 

Moreover, autophagy can contribute to therapeutic resistance by promoting cell survival upon cellular stresses induced by various forms of cancer therapies [202]. The MiT/TFE family of transcription factors are often dysregulated in several types of malignancies, including renal cancer, kidney cancer, pancreatic cancer, and malignant melanoma [203,204,205,206,207,208]. Normally, TFEB shuttling between the cytoplasm and the nucleus is regulated by nutrient availability, where starvation induces nuclear translocation [63,67]. However, chromosomal aberrations that affects TFEB localization, or mutations causing a disconnection of TFEB from proteins controlling its cytosolic retention, trigger constituent nuclear localization and promote autophagy. This is associated with tumorigenesis and more aggressive diseases [204,209,210]. 

Autophagy is also involved in the epithelial to mesenchymal transition (EMT), a critical step during metastasis where epithelial cells acquire more mesenchymal properties to increase their migratory and invasive capacity. During EMT, cells lose their polarity and cell-to-cell contacts, rearrange their cytoskeleton, and upregulate cell-to-matrix adhesions [211]. While autophagy can have a negative impact on EMT by downregulating important transcription factors in the early phases of disease [212], it can contribute to the process and enhance the invasive phenotype of cancer cells as well [213]. For example, lysosomal degradation of adhesion molecules, such as E-cadherin stimulates EMT, and autophagic recycling provides energy and nutrients to promote cell survival during the EMT-process [214,215,216]. 

### 4.2. Tumor-Induced Regulation of pH

Cancer cells often increase their glucose uptake, which promote survival and cancer progression [217]. This is often combined with an altered metabolism, where glucose molecules are fermented into lactate even in the presence of oxygen and functional mitochondria, a process commonly known as the Warburg effect [218]. Glycolysis is not as efficient as mitochondrial respiration when calculating ATP yield per glucose molecule but it provides important carbon moieties required for long-term cell growth, survival, and metastatic progression [217]. The produced lactic acid will affect the pH in the cytoplasm and therefore must be removed to avoid cytosolic acidification [219,220]. To achieve this, cancer cells often upregulate ion transporters in the plasma membrane and can also utilize the lysosomal system. By increasing the lysosomal volume and quantity, as well as the expression and activity of lysosomal V-ATPases, cancer cells enhance their proton storage capacity to maintain intracellular pH homeostasis [221]. This is often brought by the upregulation of MiT/TFE, which increases V-ATPase transcription in several malignancies, including malignant melanoma and pancreatic ductal cancer [204,208,222]. Notably, the peripheral transport of lysosomes can be induced by lowering the cytoplasmic pH [223] and consequently, lysosomal exocytosis and the release of protons to the extracellular space reduces the intracellular proton load [221]. The subsequent acidification of the tumor microenvironment can further stimulate the relocation of lysosomes to the cell periphery and promote proton release [224], which enhances the proteolytic activity of extracellularly released hydrolases. By reducing the pH in the tumor microenvironment, cancer cells are able to stimulate EMT and modulate the tumor immune response to allow the cancer to spread. Moreover, the reduced cytoplasmic proton load causes cytosolic alkalization, which promotes tumorigenesis by allowing growth factor-independent proliferation and reduce apoptosis susceptibility [225]. 

### 4.3. Lysosomes in Drug Resistance

A major problem in cancer therapy is the development of multidrug resistance, where lysosomes are suggested to play a significant role. Many chemotherapeutic drugs are small amine-containing lipophilic or amphiphilic molecules that accumulate inside lysosomes due to the large pH difference between the lysosome (pH 4.5–5) and the cytosol (pH 7–7.5). The drugs passively diffuse across the lysosomal membrane and into the lysosomal lumen where they become protonated and retained in the acidic environment [226,227]. Several chemotherapeutic drugs are sequestered by lysosomes in this way, including cisplatin [228], sunitinib [229], doxorubicin [230], and vincristine [231]. In addition, cancer cells can induce the expression of drug transporters in the lysosomal membrane to actively pump cytotoxic drugs into the lysosome [227,232]. The ABC transporter P-glycoprotein, most commonly known to induce efflux over the plasma membrane to confer resistance against various chemotherapeutic drugs [233], can also be expressed in the lysosomal membrane and mediate drug efflux into the lysosomal lumen [234]. The same has been found for the ABC transporter A3, which induces lysosome-mediated multidrug resistance in leukemia cells [235]. The sequestration of drugs inside lysosomes prevents them from reaching their intracellular target and reduces the acidity and activity of the lysosome. To compensate for the reduced lysosomal function, the cell activates TFEB-mediated lysosomal biogenesis to produce more lysosomes. Consequently, the capacity to sequester drugs increases even more and further aggravates the chemotherapeutic resistance [226,236]. Lysosomes can also relocate to the plasma membrane and secrete the contained drugs to the extracellular environment as a mechanism to confer drug resistance. Lysosomal exocytosis in connection with treatment resistance is exemplified in several types of cancer, such as leukemia [237], ovarian cancer cells [238], and pleomorphic sarcoma [239]. Noteworthy, while the majority of studies so far have demonstrated the importance of lysosomes as mediators of drug resistance [240], there are recent studies that raise concerns about the theory of drug resistance in lysosomes [241,242].

### 4.4. Lysosomal Exocytosis and Release of Extracellular Vesicles in Cancer

Lysosomal exocytosis and the secretion of lysosomal content have been shown to facilitate tumor growth and spreading. By hijacking the lysosomal exocytosis process, cancer cells are able to acidify the tumor microenvironment, remodel the extracellular matrix, and communicate with other cells in the tumor stroma, and thereby provide optimal conditions for cancer migration, invasion, and metastasis (Figure 9) [7]. Lysosomal exocytosis is associated with more aggressive tumors and enhanced invasive and metastatic capability in several cancers [196]. Below, we will focus on the impact of lysosomal function and lysosomal exocytosis for cancer progression and refer to recent reviews for more in-depth studies of how exosomes contribute to malignancy [153,243,244]. 

#### 4.4.1. Alterations of Lysosomal Function Regulate Release of Exosomes

As mentioned before, exosomes are generated in MVEs, which are formed during endosomal maturation. The MVEs can either fuse with the plasma membrane to release their content extracellularly or continue down the endocytic road and fuse with lysosomes for cargo degradation [148]. Oncogenic alterations affecting lysosomal function and lysosomal biogenesis determine the fate of MVEs, and thus the release of exosomes. Recent research has found that the inhibition of lysosome biogenesis, or a reduced lysosomal acidification, will guide the MVEs towards the plasma membrane, resulting in the release of exosomes. This is, for example, seen in cancers harboring inactivating mutations of the phosphatase PTEN, which results in the impaired activation of TFEB [245]. PTEN deficiency causes an increased release of exosomes and facilitates cell proliferation and invasion in cholangiocarcinoma cells. Moreover, impaired lysosomal function, caused by hypoxia-induced downregulation of the lysosomal V-ATPase subunit ATP6V1A, inhibits the fusion of lysosomes with MVEs and augments the secretion of exosomes [246]. Activation of the oncogenic MEK/ERK pathway can reduce the expression of lysosomal genes and reroute MVEs to the plasma membrane, instead of promoting lysosomal degradation [247]. 

In triple-negative breast cancer cells, a reduced expression of ATP6V1A and an increased release of exosomes is caused by oncogenic downregulation of sirtuin-1, a NAD^+^-dependent deacetylase [248,249]. Besides exosome release, soluble cysteine cathepsins are secreted extracellularly. Interestingly, the combination of exosomes and secreted cathepsins promote an invasive phenotype when added to 3D cultures of breast cancer cells, which is not seen when cathepsins or exosomes are added separately [248]. It has been suggested that lysosomal exocytosis and the secretion of soluble lysosomal hydrolases can induce or enhance exosome release. In ovarian cancer cells, the exposure to hypoxic conditions increases the expression of TFEB and TRPML1, which induces lysosomal exocytosis. The lysosomes dock with the plasma membrane and secrete lysosomal hydrolases to the cell culture media, which favors STAT3 controlled, Rab27-driven exosome release [250]. Moreover, in aggressive sarcoma cells, over-sialylation of LAMP1 causes the redistribution of lysosomes to the cell periphery and induces exocytosis of both soluble lysosomal hydrolases and exosomes [239]. This results in the invasion and propagation of invasive signals and mediates drug resistance via the secretion of chemotherapeutic agents. 

#### 4.4.2. Direct Shedding of EVs with Lysosomal Origin

While there are numerous studies performed on exosomes and cancer [153,243,244], less is known about the direct participation of lysosomal exocytosis and the shedding of ectosomes with lysosomal origin. As previously mentioned, UVA irradiation causes lesions in the plasma membrane, which are repaired by lysosomal exocytosis. As a result, EVs that are larger compared to conventional exosomes are generated and released within minutes after the damage, indicating that the vesicles are lysosome-derived ectosomes. Besides the expected plasma membrane markers, the ectosomes are also positive for both soluble and membranous lysosomal proteins [190]. In malignant melanomas, cells spontaneously secrete two distinct populations of EVs, corresponding to conventional exosomes and larger plasma membrane-shedded ectosomes [251]. UVA-induced plasma membrane damage greatly enhances the ectosome fraction, which is positive for the lysosomal marker LAMP2 and contains the transforming growth factor β (TGF-β). When added to unexposed control cells, the ectosomes enhance migration by upregulating TGF-β and IL6/STAT3 signaling pathways and downregulating apoptotic signaling. Thus, malignant melanoma cells induce a direct shedding of cancer-promoting, lysosome-derived ectosomes. Several studies have demonstrated that oncogenic proteins induce lysosomal exocytosis after plasma membrane repair; although, the subsequent generation of ectosomes has not been investigated. RNF167-a, a lysosome-associated ubiquitin ligase that negatively regulates lysosomal exocytosis, is inactivated in various types of cancers. The downregulation of RNF167-a causes increased lysosomal exocytosis upon streptolysin O mediated plasma membrane damage, which can be mimicked by the Ca^2+^ ionophore ionomycin [252]. Moreover, an oncogenic EGF expression stimulates peripheral trafficking of lysosome, which is regulated by the deubiquitinase, USP17. USP17 induces the secretion of cathepsin D and promotes plasma membrane repair [253]. Since lysosome-mediated plasma membrane repair is dependent on Ca^2+^ influx, shedding of lysosome-derived ectosomes can probably occur under other conditions where intracellular Ca^2+^ levels are elevated. In line with this, upregulation of the Ca^2+^ permeable cation channels TRPML1 and TPC, detected in several types of cancer, increases the intracellular Ca^2+^ levels and stimulates lysosomal fusion with the plasma membrane [176,254,255,256]. LAMP2 is suggested to be a key regulator of EV secretion in multiple myeloma cells, where EV secretion confer drug-resistance in previously sensitive cells [257]. The downregulation of LAMP2 decreased the secretion of EVs and sensitized cells to the anti-cancer drug lenalidomide. 

#### 4.4.3. Lysosomal Positioning Determines the Malignant Phenotype 

Redistribution of lysosomes to the cell periphery is a prerequisite for lysosomal exocytosis and is important for both malignant transformation and cancer progression (Figure 9) [121,148]. During the transformation process, lysosomal proteases such as cathepsin B and D are often relocated from the perinuclear area to the cell surface [258,259]. An invasive phenotype is associated with loss of cell-to-cell contacts and degradation of the surrounding ECM, facilitated by proteases, such as matrix metalloproteinases (MMP) and plasminogen. This enables cancer cells to breach through the basement membrane and allow tumor spreading. The repositioning of lysosomes to the cell periphery and subsequent exocytosis results in the release of lysosomal cathepsins, aiding the degradation of cell-to-cell adhesions and ECM [196]. Several malignant alterations that are associated with aggressive cancer phenotypes, such as the overexpression of ErbB2 or mutant K-Ras, induces the repositioning of lysosomes to the cell periphery [260,261,262], and it has been demonstrated that lysosomes are relocated to the plasma membrane at the invasive front in tumors [263,264]. In accordance, an increased expression of lysosomal cathepsins at the invasive edge is detected in several types of cancer [265,266,267,268], and is correlated to a poor prognosis [194]. Recently, we compared lysosomal characteristics in normal melanocytes and malignant melanoma and found that peripherally located lysosomes in malignant melanoma maintain the acidic pH and protease activity. In contrast, lysosomal activity is diminished and pH is increased towards the cell periphery in normal melanocytes [197]. Since most lysosomal proteases are dependent on an acidic pH for their optimal function [5], a maintained acidity around the secreted proteases would provide a more optimal environment. This has been seen during bone resorption, where lysosomal fusion with the plasma membrane facilitates the formation of a ruffled membrane containing V-ATPases. By creating a locally acidic extracellular environment, cathepsin K-mediated bone degradation is stimulated [269]. It has been suggested that cancer cells can utilize a similar mechanism [196]. In accordance, V-ATPase located at the plasma membrane contributes to cell migration and invasion in breast cancer by reducing the extracellular pH and inducing the rearrangement of actin filaments [270,271]. In contrast, the inhibition of V-ATPase activity and the neutralization of the acidic extracellular environment diminish MMP activity and reduce metastatic behavior [222].

#### 4.4.4. Lysosomal Exocytosis Remodels the Extracellular Matrix and Activates EMT 

Secreted cathepsins can cause remodeling of the ECM, either via direct cleavage of ECM substrates or by activating other ECM-degrading proteases (Figure 9). Cathepsin B has been shown to degrade type IV collagen at the surface of breast and colorectal cancer cells via a proteolytic cascade involving MMPs and the serine protease urokinase plasminogen activator (uPA) [272,273]. The proteolytic cascade resulting in collagen degradation is increased when tumor cells are cocultured with fibroblasts, and even more augmented when monocytes are present, implicating the importance of stromal cells for the invasive ability [273]. In addition, several cysteine cathepsins, including cathepsin S and cathepsin H, induce neovascularization to facilitate cancer growth [274,275]. By utilizing stromal cells in the tumor microenvironment, malignant cells can foster its surroundings to allow cancer progression. The concerted action between cancer associated fibroblasts, immune cells, and angiogenesis facilitates invasion and metastasis [276]. Cathepsins can be supplied to the tumor by stroma cells, and in turn mediate the release of growth factors such as EGF and TGF-β to transmit oncogenic signaling to neighboring and distal sites [274,277]. TGF-β is a key regulator in tumor biology and promotes cell invasion, immune evasion, and metastatic dissemination. Cathepsin-dependent secretion of TGF-β from tumor cells induces the expression of fibroblast-activating protein-α (FAP-α) and activates stromal fibroblasts [278,279]. By inhibiting lysosomal cathepsins, FAP-α activity can be diminished to reduce malignant behavior. Moreover, extracellular TGF-β is activated by uPA, which in turn can be activated by secreted cathepsin B [280]. Interestingly, while the upregulation of both cathepsin B and cathepsin L is associated with metastasis in human melanomas, immunohistochemical analysis reveals that the two proteins are located in different cell types in the tumor [279]. While cathepsin B is predominantly expressed in melanoma cells, cathepsin L is mainly found in cancer-associated fibroblasts surrounding the invading melanoma cells. Furthermore, staining intensity increases in cells closer to the invasive front in the intradermal part of the lesion, indicating that the activation of fibroblasts and concomitant cathepsin L expression are important facilitators of malignant melanoma dissemination [279]. 

Increased expressions of lysosomal membrane proteins and cysteine cathepsins are associated with the upregulation of EMT marker proteins, induction of mesenchymal phenotype, and treatment resistance [281,282,283,284,285]. Induction of EMT is often regulated by TGF-β signaling, and inhibition of cysteine cathepsins can reverse this process and attenuate invasive growth [284,286,287]. Activation of EMT is correlated to a peripheral localization of lysosomes, augmented secretion of lysosomal proteases, and increased invasive behavior [288]. The inhibition of Arl8b or the Na^+^/H^+^ exchanger 1 (NHE1) prevents anterograde lysosomal transport, reverses EMT, and reduces malignancy. As previously mentioned, oncogenic mutations in K-Ras induces the redistribution of lysosomes to the plasma membrane [262] and recently it was shown that, upon ionizing radiation, activating K-Ras mutation stimulates the upregulation of cathepsin L and induction of EMT [283]. Together, this supports an active role for lysosomal exocytosis of lysosomal cathepsins in EMT. 

#### 4.4.5. The Lysosomal Membrane in Cancer Progression 

While cathepsins and other lysosomal proteins are known to be fundamental for cancer progression, the lysosomal membrane itself can contribute to malignant behavior as well (Figure 9). The invasive capacity of a cancer cell is dependent on its ability to breach through the basement membrane and spread to surrounding tissues. By forming invadopodia, small matrix metalloproteinase rich F-actin membranes, cancer cells can adhere to their surroundings and create forces necessary to make a small breach in the basement membrane [289]. A large invasive protrusion is then formed to widen the opening [290,291]. Studies in *C. elegans* have shown that lysosomal exocytosis is used to provide the extra membrane parts required to expand the membrane and form the large protrusion [292]. Alterations in glycosylation patterns of lysosomal membrane proteins can also increase the metastatic ability of cancer cells by serving as ligands for extracellularly located selectins and galectins [263,293]. 

#### 4.4.6. Regulation of Cancer-Induced Lysosomal Repositioning

Lysosomal repositioning can be induced by several factors in the tumor microenvironment, such as low pH, as discussed above. Growth factors affect lysosomal localization and both the hepatocyte growth factor (HGF) and epithelial growth factor (EGF) induce a peripheral localization of lysosomes and secretion of lysosomal proteases [294,295]. As previously mentioned, lysosomal exocytosis is induced by elevated levels of intracellular Ca^2+^, and several lysosomal cation channels are dysregulated in cancer. A high expression of TRMPL1 is associated with a poor prognosis in pancreatic ductal adenocarcinoma [255] and by inhibiting its function, the growth of tumor cells is reduced [254,296]. Moreover, by disrupting the lysosomal TPC, cells are unable to form leading edges, a prerequisite for cell migration [256]. 

Since lysosomal exocytosis is dependent on vesicular transport between the cell center and the cell periphery, it is not surprising that proteins involved in lysosomal trafficking often are deregulated in cancer. The loss of the microtubule-associated protein 1 light chain 3 gamma (LC3C) results in a peripheral redistribution of lysosomes, enhanced exocytosis, and more aggressive tumors [297]. Moreover, an altered expression of proteins regulating anterograde and retrograde transport of lysosomes is found in various cancers. For example, the kinesin-1 protein KIF5B is upregulated in highly invasive MCF7 breast cancer cells expressing constitutively active ErbB2, as compared to non-mutated MCF7-cells [298]. The association of lysosomes to kinesins is mediated by Arl8b and the BORC complex, and a high expression of both Arl8b and the BORC-subunit proteins, BLOC1S2 and BORCS5, are correlated to a poor prognosis and an increased invasive capacity in breast cancer and prostate cancer, respectively [295,299]. Furthermore, low expression of Rab7 and its effector RILP is correlated to malignant behavior, as it reduces the number of juxtanuclear lysosomes and promotes lysosomal exocytosis and metastatic spreading [300,301,302]. In a histological study of melanocytic lesions, it was found that tumors tune their Rab7 expression to control the malignant transformation. Rab7 expression is considerably higher in melanoma in situ, where the tumor grows in a radial pattern in the epidermis and stimulates cell proliferation. However, the expression is reduced when the tumor adopts a vertical growth pattern and invades the dermis. Finally, Rab7 is upregulated again at distal metastatic locations [301]. Several other Rab proteins, such as Rab25, Rab26, Rab27, and Rab38, that stimulate anterograde lysosomal transport are upregulated in various forms of cancer [7,303], while Rabs responsible for retrograde transport are found to be downregulated. 

Lysosomal membrane proteins, including the lysosomal membrane proteins LAMP1 and LAMP2, are upregulated during malignant transformation [304] and it has been shown that LAMP1 is required for docking at the plasma membrane upon lysosomal exocytosis [239,305]. NEU1, a sialidase that removes sialic acids from LAMP1, reduces lysosomal exocytosis. By downregulating NEU1, tumor cells can enhance the release of lysosomal hydrolases and exosomes, which enhances invasive properties such as matrix degradation, propagation of invasive signals, and efflux of lysosomotropic chemotherapeutics [34,239]. In pleomorphic and metastatic sarcomas, the downregulation of NEU1 correlates to an increased expression and interaction between LAMP1 and myosin-11, a motor myosin protein, which facilitates lysosome trafficking to the cell periphery and lysosomal exocytosis [239]. An increased expression of the LAMP-family protein LAMP5, as well as the lysosomal integral protein LIMP2, has also been associated with cancer progression and metastasis in various cancers [306,307,308]. Although less studied compared to LAMP1 and LAMP2, LAMP5 overexpression is associated with lysosome repositioning to the cell periphery in mixed lineage leukemia [309], and the promotion of cancer stemness and EMT in gastric cancer [282]. By targeting LAMP5 expressed on the cell surface, it is possible to reduce cell viability, which indicates a potential use as a therapeutic target in blood cancers [309]. TMEM106B is another single-pass lysosomal transmembrane protein that was recently shown upregulated in human lung adenocarcinomas [310]. The protein functions as a driver of invasion and metastasis by inducing TFEB-mediated synthesis of lysosomal proteins and subsequent Ca^2+^-dependent exocytosis of lysosomal cathepsins. 

### 4.5. Induction of Lysosomal Damage as a Therapeutic Target

While lysosomes are clearly implicated in tumor progression and drug resistance, tumorigenic changes of lysosomal properties can also sensitize cancer cells to lysosomal damage (Figure 10) [77,197,304,311]. Recently, it has been recognized that lysosomal stability depends on its intracellular localization. As mentioned above, anterograde, peripheral transport of lysosomes is mediated by kinesins, and knockdown of several different kinesins alters lysosomal function, destabilizes the lysosomal membrane and sensitizes cells to lysosome-disrupting drugs [312]. By upregulating Rab7 or inhibiting the association between SKIP and kinesin-1, lysosomes are relocated to the perinuclear area, which sensitizes malignant melanoma cells to the lysosome-destabilizing drug L-leucyl-L-leucine methyl ester [197]. Conversely, by stimulating peripheral localization via Rab7 downregulation, lysosomes are less sensitive to damage and cell death is reduced. Overexpression of HER2, found in 15–30% of all breast cancers, is associated with metastasis and poor prognosis. Targeting of HER2 has greatly increased patient survival in breast cancer, but development of drug resistance is still a major therapeutic obstacle [313]. A recent study by Hansen et al. showed that overexpression of HER2 was associated with invasion-promoting peripheral localization of lysosomes [314]. Interestingly, by performing a drug screen, the authors identified several drugs that were able to relocate lysosomes from the cell periphery to the perinuclear area, which reverted the invasive phenotype and induced lysosome-dependent cell death in HER2 positive breast cancer cells. 

Drug resistance is a major challenge in cancer therapy. Therefore, the use of lysosomotropic detergents to destabilize the lysosomal membrane and induce lysosome-dependent cell death is a promising strategy to target treatment-resistant cancer [195]. As previously mentioned, lysosomotropic agents cross the lysosomal membrane as uncharged molecules and become protonated inside the acidic environment [315]. The resulting accumulation can eventually destabilize the lysosomal membrane and induce LMP. Although some lysosomotropic detergents might solubilize the lysosomal membrane directly [316], the cause of membrane damage is dependent on their chemical structure [80]. Lysosomes contain intraluminal membranes where most of the lipid metabolism occur. The membranes are negatively charged, due to the presence of the phospholipid Bis(monoacyl)glycerophosphate (BMP), to enable access of positively charged lysosomal lipases [317]. Inhibition of acid sphingomyelinase (ASMase), the lipase responsible for the conversion of sphingomyelin to ceramide, results in accumulation of sphingomyelin, which alters the lipid composition and destabilizes the lysosomal membrane [77,318,319]. Cancer cells often show an altered lysosomal membrane lipid composition and display low levels of sphingomyelin, which sensitizes them to sphingomyelin accumulation [77,320,321]. Reduced mRNA levels of SMPD1, the gene encoding for ASMase, is found in several types of cancer, including liver cancer, renal cancer and head and neck carcinomas [195]. However, the mRNA levels do not necessarily represent protein activity since the interaction between ASMase and BMP is stabilized by the stress-activated heat-shock protein HSP70, which is overexpressed in various forms of cancer [319,322]. By targeting HSP70, ASMase activity can be diminished, increasing the buildup of sphingomyelin and compromising lysosomal integrity [323,324]. Several drugs, both experimental and clinically approved, show anticancer effects by inhibiting ASMase and other lysosomal lipases, thereby increasing sphingomyelin levels [77,195]. 

One such group of drugs are the cationic amphiphilic drugs (CADs), which exhibit lysosomotropic properties and have emerged as promising candidates to reduce treatment resistance in cancer therapy [325]. These drugs incorporate into the intraluminal membranes, neutralize the negative charge of BMP and inhibit the activity of several lipases, including ASMase [326]. A wide variety of drugs have CAD-like properties and are used clinically to treat conditions such as allergy, heart arrythmia and psychiatric disorders [327,328]. More than 30 different CADs have been shown to exert anti-tumorigenic effects [326] and several can revert multidrug resistance by destabilizing the lysosomal membrane [329,330,331]. A Danish cohort study including patients with non-localized cancer demonstrated that concomitant use of CAD antihistamines and standard chemotherapeutic drugs significantly reduce patient mortality rate [332]. Moreover, the use of lysosome-specific nanoparticles that induce lysosomal membrane permeabilization is emerging as a potential therapeutic approach [333]. The nanoparticles can induce LMP directly or act as carriers of chemotherapeutic drugs [334,335]. A recent study demonstrated that the combined use of CADs with nanoparticles carrying siRNA significantly improves gene silencing due to increased permeability of the lysosomal membrane [336].

## 5. Conclusions

The 70-year-long story of the lysosome that started as a simple waste bag for degradation has now evolved into the appreciation of lysosomes as central organelles for cellular homeostasis, nutrient sensing, and regulator of cell death and survival. In this review, we summarized the normal function of lysosomes and the cancer-associated changes that promote the progression of malignant disease. As a therapeutic target, the lysosome has not yet reached its full potential, and lysosomal positioning is a promising area of interest that requires further research. Emerging evidence has established a correlation between lysosomal intracellular position and function, and its importance for malignancy. TFEB activity is central for controlling lysosomal biogenesis and TFEB-dependent upregulation of Ca^2+^-channels in the lysosomal membrane determines cytosolic Ca^2+^ level, which in turn controls lysosomal exocytosis and the release of ectosomes. In addition, reduced lysosomal function or TFEB downregulation stimulates exosome secretion from MVEs. Studies of the signaling ability of extracellular vesicles and their impact on intercellular crosstalk to modulate tumor microenvironment and facilitate metastasis is still in its infancy. Compared to the interest devoted to the role of MVE-originating exosomes, lysosomal exocytosis with subsequent generation of ectosomes with lysosomal origin is still underdeveloped. Thus, targeting the intracellular position of the lysosome is an important strategy to control the ability of exocytosis, which prevents not only remodeling of the extracellular matrix, but also affects cell migration and EMT-favoring properties. Exploiting the knowledge that perinuclear lysosomes have reduced membrane stability is a central tool for amplifying different modes of programmed cell death. With this direction for future research, we might reach new therapeutic strategies to potentiate cancer treatment.

## Figures and Tables

**Figure 1 cells-13-00459-f001:**
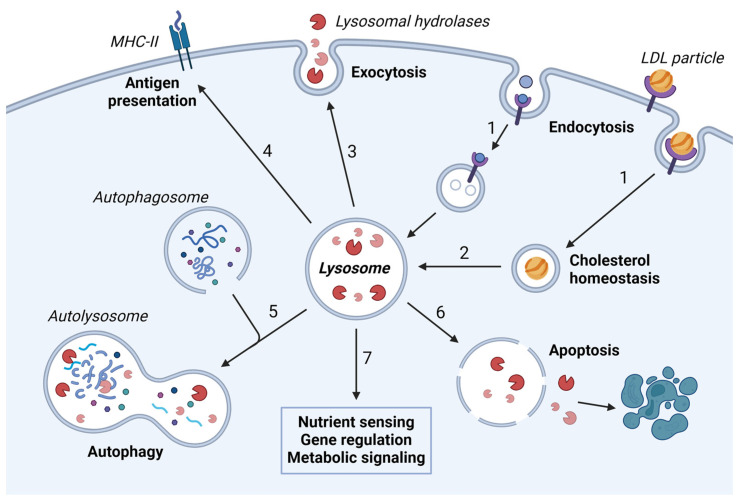
Lysosomal function. Lysosomes receive extracellular cargo via receptor-mediated endocytosis and uptake of bulk material via pinocytosis and phagocytosis (1). By utilizing receptor-mediated endocytic uptake of LDL particles, lysosomes participate in cholesterol homeostasis (2). Translocation of lysosomes to the plasma membrane and exocytosis of hydrolytic enzymes (3). mediates e.g., bone remodeling, degradation of the extracellular matrix and cell-to-cell communication. Lysosomal exocytosis is also important for plasma membrane repair where the lysosome donates its membrane to repair the lesion (3). Lysosomal processing of foreign and endogenous material allow antigen presentation on MHC-II molecules (4). Intracellular material is degraded in autolysosomes, formed by the fusion of a lysosome and an autophagosome (5). Damage to the lysosomal membrane results in release of lysosomal proteases to the cytosol and cell death induction (6). By acting as a central hub for nutrient sensing, the lysosome is involved in the regulation of gene expression and metabolic signaling (7). Image created with BioRender.com, adapted from [8].

**Figure 2 cells-13-00459-f002:**
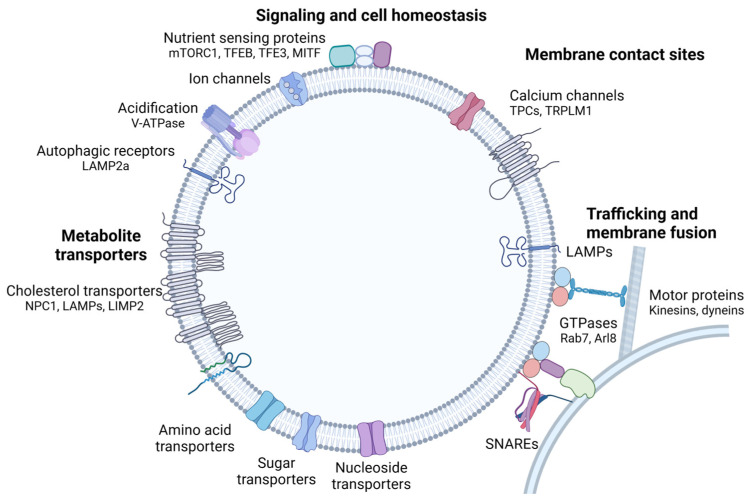
Function of the major lysosomal membrane proteins. Lysosomal membrane proteins play crucial roles in maintaining the function, structure, and integrity of the organelle. They are involved in key processes such as metabolite transport, which includes proton pumping and acidification. Moreover, nutrient-sensing membrane proteins are involved in cell signaling and regulation of homeostasis and metabolism. Membrane contact sites coordinate with, for example lipid metabolism and Ca^2+^ signaling, while proteins associated with the membrane also regulate the dynamics of lysosomal fusion and fission, as well as lysosome trafficking along microtubule. Image created with BioRender.com, first published in [8].

**Figure 3 cells-13-00459-f003:**
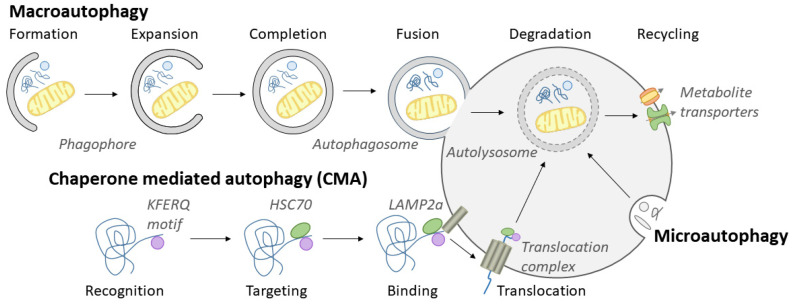
Autophagic pathways. Three main routes of autophagy are identified. Large cytoplasmic material is mainly degraded via macroautophagy, where the material is sequestered by a phagophore, forming an autophagosome. The autophagosome fuses with a lysosome to form an autolysosome in which degradation takes place. Cytosolic proteins are degraded by chaperone-mediated autophagy (CMA), where the chaperone protein HSC70 recognizes a target motif on cytosolic proteins and facilitates its binding to the CMA-receptor, LAMP2a. The binding induces LAMP2a oligomerization and allows translocation of the target protein to the lysosomal lumen. During microautophagy, invaginations are formed in the lysosomal membrane to allow a direct uptake of cytoplasmic proteins and smaller structures into the lysosome. Image first published in [8].

**Figure 4 cells-13-00459-f004:**
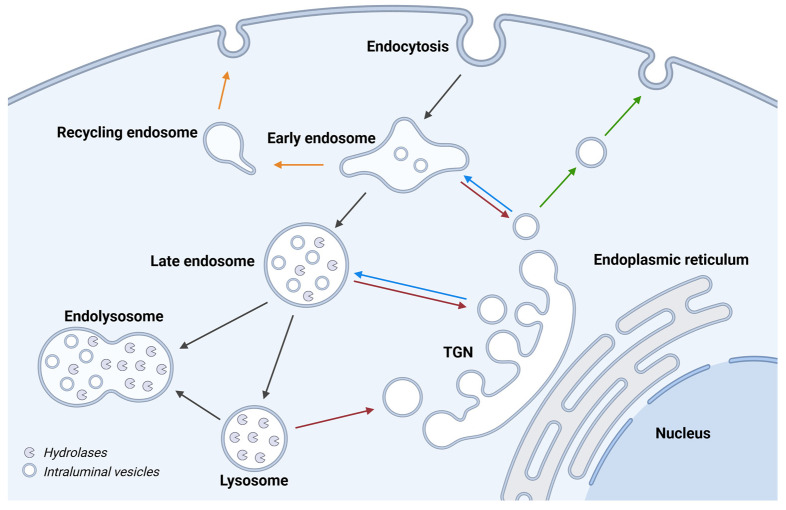
The endocytic pathway. Soluble and membrane-bound material are taken up via endocytosis and sorted in early endosomes, where the majority is recycled back to the plasma membrane via recycling endosomes (orange arrows). Material to be degraded follows the endocytic route to the lysosome (grey arrows). During this process, delivery of newly synthesized lysosomal components from the trans-Golgi network (TGN) allows maturation of early endosomes into late endosomes and lysosomes. The delivery from TGN can occur via the secretory pathway (green arrows) where secreted components are taken up via endocytosis, or via a direct fusion of Golgi-derived vesicles with early and late endosomes (blue arrows). Endosomal maturation also includes accumulation of intraluminal vesicles to allow sorting and degradation of transmembrane cargo. Via retrograde transport, TGN-specific material is recycled from the endolysosomal compartments (red arrows). Transient and complete fusion events between endosomes and lysosomes generates endolysosomes and facilitates exchange of material and cargo degradation. Image created with BioRender.com, adapted from [8].

**Figure 5 cells-13-00459-f005:**
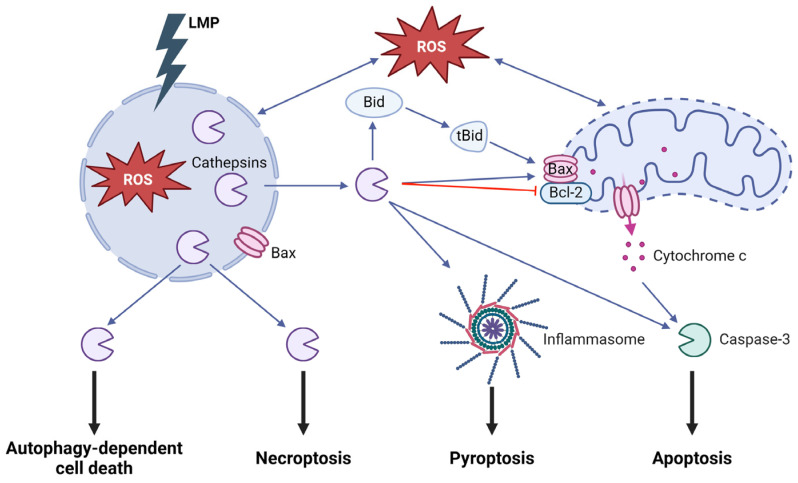
Participation of cathepsins in regulated cell death. Lysosomal membrane permeabilization (LMP) results in release of cathepsins to the cytosol and is associated with increased reactive oxygen species (ROS). Hyperactivation of autophagy during e.g., mitophagy and ER-phagy results in altered lipid metabolism and lysosomal membrane destabilization with ensuing cathepsin release and autophagy-dependent cell death. LMP and subsequent release of cathepsins can trigger necroptosis, a specific form of cell death with necrosis like morphology. Pyroptosis is the consequence of inflammatory processes where cathepsin-induced assembly of the inflammasome activates caspase-1. Cytosolic cathepsins can also induce cytochrome c release from the mitochondria to activate the intrinsic pathway to apoptosis. This is mediated by proteolytic activation of Bid or inactivation of anti-apoptotic Bcl-2 proteins, or a direct proteolytic processing of caspases. Mitochondrial outer membrane permeabilization can further amplify lysosomal damage by causing elevated levels of oxidative stress, and by inducing Bax oligomerization in the lysosomal membrane. Image created with BioRender.com.

**Figure 6 cells-13-00459-f006:**
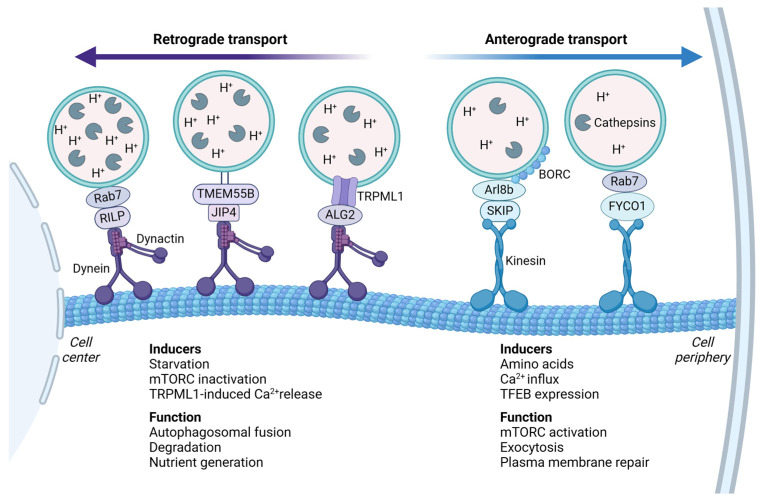
Regulation of lysosomal transport. Retrograde transport towards the cell nucleus is mainly orchestrated by the dynein/dynactin motor protein complex. The GTPase Rab7 mediates lysosomal coupling to the dynein/dynactin complex with the aid of its effector RILP. Alternatively, TRPML1 mediated Ca^2+^ release, or starvation-induced transcription of the lysosomal transmembrane protein TMEM55B, induce the interaction with the adaptor proteins ALG2 and JIP4, respectively, to couple lysosomes to the dynein/dynactin complex. Anterograde movement to the cell periphery is instead mediated by kinesin proteins. The assembly of BORC, Arl8b and SKIP, or Rab7 and FYCO1, links lysosomes to microtubule. The transport is regulated by nutrient availability, TFEB activity and Ca^2+^ levels. Lysosomes adjacent to the nucleus are acidic and proteolytically active, and fuse with autophagosomes to allow degradation and nutrient generation. Peripherally located lysosomes are involved in lysosomal exocytosis and plasma membrane repair and induce mTORC activation. Image created with BioRender.com.

**Figure 7 cells-13-00459-f007:**
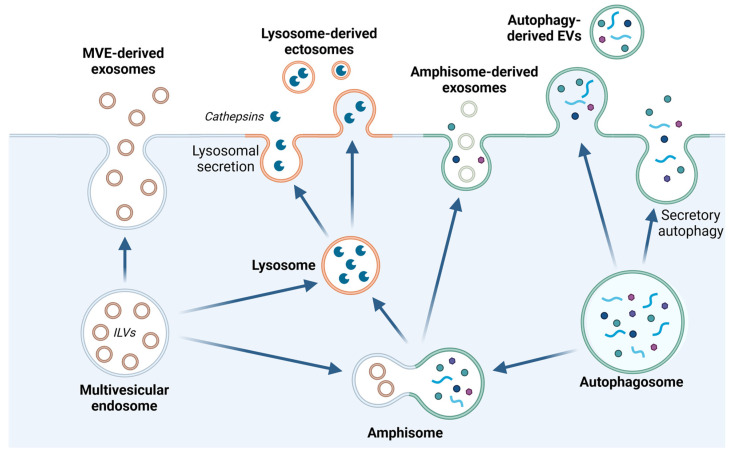
Routes of secretion from the endolysosomal system. Intraluminal vesicles (ILVs) originating from multivesicular endosomes (MVEs) are secreted as exosomes. Mature lysosomes secrete soluble proteins via lysosomal exocytosis or as ectosomes following fusion with the plasma membrane. While autophagosomes normally fuse with lysosomes to allow degradation of their cargo, they can also reroute to the plasma membrane and release soluble content or extracellular vesicles (EVs). Furthermore, autophagosome fusion with multivesicular endosomes forms a hybrid organelle termed amphisome, which can release autophagic degradation products and exosomes of both endosomal and autophagic origin. Image created with BioRender.com.

**Figure 8 cells-13-00459-f008:**
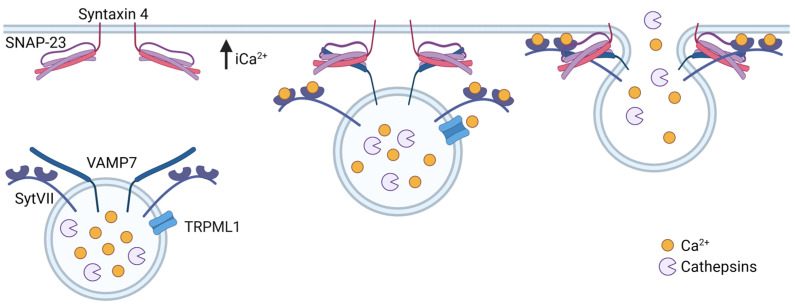
Lysosomal exocytosis. Lysosomal fusion with the plasma membrane is triggered by increased intracellular Ca^2+^ (iCa^2+^), originating from intracellular Ca^2+^ stores or via influx from the extracellular space. Ca^2+^ binds to and activates synaptotagmin VII (SytVII), resulting in transfer of lysosomes to the plasma membrane. After tethering to the plasma membrane, docking and merging of the phospholipid bilayers is performed by interaction between VAMP7, which is a lysosomal v-SNARE, and the t-SNAREs SNAP-23 and syntaxin 4 on the plasma membrane. Upon membrane fusion, the lysosomal content is released extracellularly. Image created with BioRender.com.

**Figure 9 cells-13-00459-f009:**
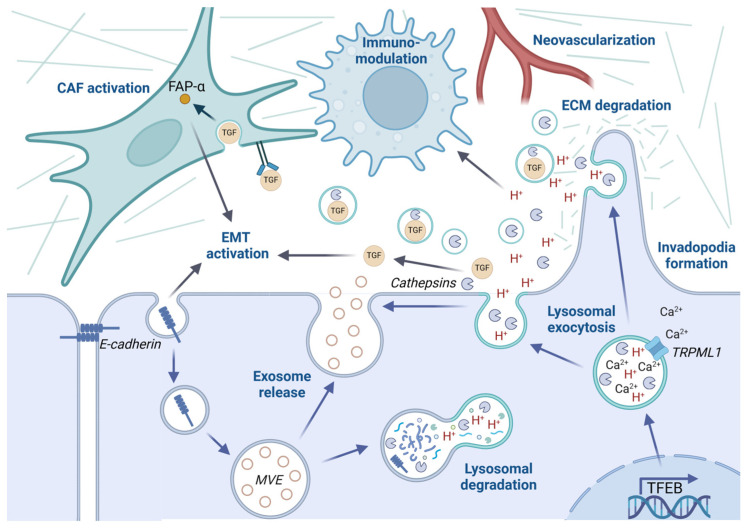
Cancer-promoting effects of lysosomal exocytosis. Several cancer-associated changes, such as TFEB upregulation and increased expression of Ca^2+^ permeable channels, increase lysosomal exocytosis and enhance release of both soluble content and ectosomes shedded from the plasma membrane. The released lysosomal content has been shown to mediate extracellular matrix (ECM) degradation, immunomodulation, and neovascularization. Lysosomal exocytosis induces TGF-β signaling to activate cancer-associated fibroblasts (CAFs) and promotes epithelial to mesenchymal transition (EMT). EMT is further stimulated by downregulation of adhesion molecules such as E-cadherin, facilitated through lysosomal degradation. Secreted cathepsins can promote release of exosomes from MVEs to further modulate the tumor microenvironment. By utilizing lysosomal membrane fusion with the plasma membrane, the cancer cell can elongate forming invadopodia to create breaches in the basement membrane and facilitate tumor invasion. Image created with BioRender.com.

**Figure 10 cells-13-00459-f010:**
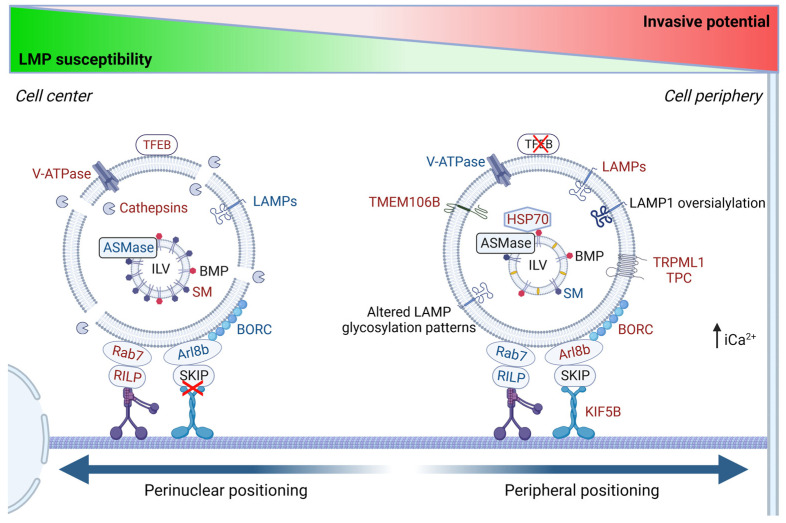
Cancer-associated changes affecting lysosomal positioning. Peripheral positioning of lysosomes is Ca^2+^ dependent and increases lysosomal exocytosis and tumor invasiveness. Contrarily, tumor-induced perinuclear positioning is often associated with increased susceptibility to lysosomal membrane destabilization and lysosome-induced cell death. Lysosomal localization and membrane stability is determined by a variety of upregulated (red) and downregulated (blue) genes, including motor proteins and lysosomal membrane proteins. Cancer cells have relatively low levels of sphingomyelin (SM) compared to normal cells. SM is converted to ceramide on intraluminal vesicles (ILVs) in the lysosome by acid sphingomyelinase (ASMase). The action of ASMase is mediated by the negatively charged lipid BMP, an interaction that is stabilized by HSP70. Accumulation of SM causes lysosomal membrane permeabilization and cathepsin-induced cell death. Image created with BioRender.com.

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
