# Peer review of "Lysosomes in Cancer—At the Crossroad of Good and Evil"

_cells, 2024, doi:10.3390/cells13050459_

Round 1
Reviewer 1 Report
Comments and Suggestions for Authors
In my opinion, the article entitled Lysosomes in cancer – at the crossroad of good and evil is a great work, and it is suitable for publication in Cells. The review is very well documented, and the text is properly written, I have enjoyed reading it. Figures are clear and very well displayed. The number of References is appropriate, and they are up to date. English is proper and I only have a few suggestions:
1.- Lines 273: It is true that most of the lysosome transport is orchestrated by dyneins for retrograde transport and kinesins for anterograde movement buts there are other possibilities. I only add a suggestion:
Vesicle-associated motors are of two types depending on the direction in which the vesicle is transported: plus-end-directed motor proteins (N-kinesins) that transport vesicles toward the cellular periphery and minus-end directed motor proteins (dynein and members of the C-kinesin family such as KIFC2 and KIFC3) that move vesicles to the perinuclear area: Bejarano et al. Aging Cell. 2018;17: e12777.
https://doi.org/10.1111/acel.12777
2.- Line 306: Regarding to lysosome retrograde transport I would like to remark that there are other proteins, only a suggestion:
Taken together, these results indicate that KIFC3 acts as the major minus-end-directed motor protein required for traffic of lysosomes toward the perinuclear region for their fusion with autophagosomes, and that by reducing levels of KIFC3 to mimic our observations in primary fibroblasts from old animals, it is possible to recapitulate the changes in lysosomal motility and autophagy observed in aging: Bejarano et al. Aging Cell. 2018;17: e12777.
https://doi.org/10.1111/acel.12777
3.- Line 506: This could be two useful references:
GSK-3β signaling determines autophagy activation in the breast tumor cell line MCF7 and inclusion formation in the non-tumor cell line MCF10A in response to proteasome inhibition.
DOI: 10.1038/cddis.2013.95
Breast cancer cell line MCF7 escapes from G1/S arrest induced by proteasome inhibition through a GSK-3β dependent mechanism.
DOI: 10.1038/srep10027
Author Response
We thank the reviewer for the kind words and the valuable suggestions of references. We have included reference from Bejarano et al about kinesin in minus-end directed transport in section 2.1 (line 422-424). We have also included one of the suggested references regarding proteasome inhibition in section 4.1 (line 600, and reference 198).
Reviewer 2 Report
Comments and Suggestions for Authors
This manuscript discusses the intricate role of lysosomes in cancer biology. The authors explore how lysosomes have emerged as critical organelles involved in nutrient sensing, intracellular signaling, programmed cell death, and supporting the growth of cancer cells. The authors discuss the dynamic nature of lysosomes, their fusion and fission events that facilitate intercellular communication, and their role in exocytosis, releasing lysosomal content and extracellular vesicles, which play a significant role in cancer progression. Furthermore, the manuscript also discusses the delicate balance between lysosomal function and susceptibility to lysosomal membrane permeabilization, providing an alternative approach to induce cell death in cancer cells. The authors highlight the importance of lysosomal structure, intracellular positioning, and transport regulation within the cell. This manuscript offers a comprehensive overview of the multifaceted functions of lysosomes in cancer, emphasizing their potential as therapeutic targets to improve cancer treatment. Here are review points that could further enhance the comprehension of this manuscript using Figure as a review section.
Figure 1, please itemize the components and organelles and mark the consequence (arrow) as enhanced or decreased, and discuss this in the main text.
Figure 3 shows that the fusion between autophagosome and autolysosome could be connected without an inner member. There is a CMA abbreviation in the legend but not in the figure. Hsc70 is HSC70. How HSC70 works on autophagy could be further discussed.
Figure 4 is confusing; the organelles are not easy to recognize. What are these circles? The objectives should be clearly labeled. What do these bi-direction arrow indicates?
Figure 6, please label the extracellular or intracellular; what does both sides’ membrane indicate? The + symbol could be replaced by other symbols since there is an H+ inside the lysosome. What are the two different colors and sizes inside the lysosome? Where are the LAMP1 and LMAP localized, as described in lines 326 and 327? The authors should discuss the mechanism that leads lysosomes toward the retrograde or anterograde direction.
Figure 7 shows there are different components inside different -somes. Please indicate their roles.
Figure 9, how these proteases are involved in protein degradation and autophagy and how cancer cells hijack these processes for their benefit, especially intracellular signaling transduction, should be further discussed.
Reviewer 3 Report
Comments and Suggestions for Authors
The authors that summarized the lysosome function linked many roles within and out of cells such as degradation, recycling and regulating cell functions. Here, authors that more propose lysosome function on potential cancer treatment that it is interesting, but should be enhanced on this.
Some comments as following:
1. What kind of cancer cells are involved in should be mention for cancer progression
2. How to induce the cell death within cells through lysosome-mediated that is key point on this review should be figure out a summary figure on this.
Reviewer 4 Report
Comments and Suggestions for Authors
Major points:
1. Section Introduction to lysosome. This paragraph is rather superficial and inaccurate. In the beginning, lysosomes were understood as static organelles involved in cellular catabolism and cell death, as you yourself state later in the text (paragraph 1.4.). Currently, lysosomes are understood as dynamic organelles involved in (here you can name dozens of processes and / or cellular functions).
2. Paragraph 1.1. You are dealing with only one type of hydrolases, namely cathepsins. The others are not mentioned even in outline. On the other hand, the classification of cathepsins is insufficient. Given that you deal with cathepsins throughout the text you should provide more detail information about these proteases. For example, the fact that some of them can be active around neutral pH should be also mentioned. Recent paper on this issue by Yadati et al., 2020 can help you.
3. Paragraph 1.1.1. You are dealing with only two types of lysosomal membrane proteins: LAMPs and v-ATPase. Ion channels and other transporters, including their basic functions, are not mentioned. An enumeration of channels and transporters for Ca2+ is mentioned in the following paragraph. Unfortunately, it is incomplete. Please, see Wu et al., Cancers (Basel) 2021, 15;13(6):1299.
4. I'm missing the section dealing with the origin of lysosomes.
5. Paragraph 1.3. Unfortunately, the authors only mention one TFEB activator. Please, see Ballabio and Bonifacino, Nat Rev Mol Cell Biol. 2020, 21(2):101-118.
6. Paragraph 1.4. Please note that lysosomes are involved in two types of regulated cell death (RCD): autophagy-dependent cell death (which you didn't mention) and lysosome-dependent cell death (Galluzzi et al., Cell Death Differ. 2018, 25(3):486-541). Unfortunately, you limited yourself to lysosome-dependent cell death, which is initiated by permeabilization of the lysosomal membrane (LMP) and can manifest itself differently depending on the extent of permeabilization and other circumstances (Aits and Jäättelä J Cell Sci. 2013, 126(Pt 9):1905-1912; Repnik et al., 2014, Mitochondrion. 2014, 19 Pt A:49-57; Wang etal., 2018, Traffic. 2018, 19(12):918-931; Galluzzi et al., Cell Death Differ. 2018, 25(3):486-541 ).
The term programmed cell death has been abandoned and is replaced by the term RCD (Galluzzi et al., Cell Death Differ. 2018, 25(3):486-541). The term necrosis has been abandoned, the concept of programmed necrosis does not exist anyway. According to Galluzzi et al., 2018, which you refer to, there is only MTP-driven necrosis and Necroptosis. What is still used is necrotic or necrotic-like morphology referring to morphological features of dying cells... In naming the individual types of regulated cell death sub-routes, please be consistent with the Nomenclature Committee for Cell Death.
7. Paragraph 4.3. There is no evidence at all that lysosomal sequestration of weak-base drugs is a clinically relevant mechanism of resistance.
Reversible binding and sequestration are important and need to be taken into account in the static in vitro situation. However, there is a fundamental difference between the dynamic in vivo and in vitro situation. Reversible binding and sequestration are irrelevant under steady state serial dosing conditions (which applies to almost all oncology pharmaceutical interventions) and cannot be mechanisms of resistance. In this situation, the extracellular free drug concentration is controlled by intrinsic clearance which in turn controls the intracellular drug concentration, influenced also by the membrane permeability of the drug and the presence of transporter systems (Smith et al., 2010, Nat. Rev. Drug Discov. 9(12), 929-939; Smith and Rowland, 2019, Drug Metab. Dispos. 47(6), 665-672).
Lysosomal sequestration of weak-base drugs as a mechanism of resistance has not been clearly demonstrated even in in vitro experiments (Mlejnek, Cells. 2023, 12(5):709; Mlejnek, Pharmacol Res. 2024, 199:107025).
8. Paragraph 4.4.6. line 819. Considering what I have written above and substantiated with citations, lysosomal-mediated drug resistance can hardly be a major challenge in cancer treatment. Lysosomes undoubtedly play a crucial role in many (if not all) aspects of cancer, but lysosome-mediated resistance is most certainly not it. However, they influence drug resistance through autophagy, as you correctly write in paragraph 4.1. and through affecting tumor microenvironment…
Round 2
Reviewer 4 Report
Comments and Suggestions for Authors
Thank you for the detailed replies to my comments and for editing the text.
Now I have no comments on your text.
I wish you the best of luck in your future work!
Author Response
We thank the reviwer for these kind words.